# Plug-and-Play Stability for Intracortical Brain-Computer Interfaces: A One-Year Demonstration of Seamless Brain-to-Text Communication

**Chaofei Fan**[6, †], **Nick Hahn**[3], **Foram Kamdar**[3], **Donald Avansino**[8], **Guy H. Wilson**[2], **Leigh Hochberg**[10,11,12], **Krishna V. Shenoy**[1,4,5,7,8,9], **Jaimie M. Henderson**[3,7, ‡], and **Francis R. Willett**[8, ‡]

[1]Bio-X Program, Stanford University
[2]Department of Neuroscience, Stanford University
[3]Department of Neurosurgery, Stanford University
[4]Department of Neurobiology, Stanford University
[5]Department of Bioengineering, Stanford University
[6]Department of Computer Science, Stanford University
[7]Wu Tsai Neurosciences Institute, Stanford University
[8]Howard Hughes Medical Institute at Stanford University
[9]Department of Electrical Engineering, Stanford University
[10]School of Engineering and Carney Institute for Brain Science, Brown University
[11]VA RR&D Center for Neurorestoration and Neurotechnology, VA Providence Healthcare System
[12]Center for Neurotechnology and Neurorecovery, Department of Neurology, Massachusetts General Hospital, Harvard Medical School
[†]Correspondence: stfan@stanford.edu
[‡]Equal contribution

## Abstract

Intracortical brain-computer interfaces (iBCIs) have shown promise for restoring rapid communication to people with neurological disorders such as amyotrophic lateral sclerosis (ALS). However, to maintain high performance over time, iBCIs typically need frequent recalibration to combat changes in the neural recordings that accrue over days. This requires iBCI users to stop using the iBCI and engage in supervised data collection, making the iBCI system hard to use. In this paper, we propose a method that enables self-recalibration of communication iBCIs without interrupting the user. Our method leverages large language models (LMs) to automatically correct errors in iBCI outputs. The self-recalibration process uses these corrected outputs ("pseudo-labels") to continually update the iBCI decoder online. Over a period of more than one year (403 days), we evaluated our Continual Online Recalibration with Pseudo-labels (CORP) framework with one clinical trial participant. CORP achieved a stable decoding accuracy of $93.84\%$ in an online handwriting iBCI task, significantly outperforming other baseline methods. Notably, this is the longest-running iBCI stability demonstration involving a human participant. Our results provide the first evidence for long-term stabilization of a plug-and-play, high-performance communication iBCI, addressing a major barrier for the clinical translation of iBCIs.

37th Conference on Neural Information Processing Systems (NeurIPS 2023).

# 1 Introduction

The ability to communicate with others is a ubiquitous, everyday function. Neurological disorders such as amyotrophic lateral sclerosis (ALS) can often lead to loss of speech. Restoring lost communication can "improve relationships, increase participation in family and community life, offer a greater sense of independence and help a patient make important medical decisions" [4]. Noninvasive assistive technologies such as eye-tracking and electroencephalography (EEG) have helped reestablish communication channels with patients but suffer from low information transfer rates [29]. More recently, progress in intracortical brain-computer interfaces (iBCIs) that record neural activity at single-neuron resolution offers a promising path to restore rapid communication [30, 47]. For example, [47] demonstrated a high-performance iBCI that can accurately decode attempted handwriting movements into text from neural activity recorded with microelectrode arrays placed in the motor cortex.

Despite these advances, numerous challenges must be addressed before iBCIs can be effectively utilized by those who require them. One key challenge is to maintain accuracy in the face of nonstationary neural recordings. iBCIs often require daily recalibration to maintain accuracy. During the recalibration process, users must cease using the iBCIs for some length of time, diminishing the advantages of high-performance iBCIs. Nonstationarities are caused by various factors, including microelectrode array movement [40], device degradation [50], single neuron physiological changes [33, 9, 3], and variability at the behavioral level [7, 46]. These changes occur across different timescales, with single and multi-unit baseline firing rates changing within minutes [22, 33] and response selectivity often varying over days [7]. As these changes accumulate, an iBCI decoder fit to a specific time period gradually becomes outdated, resulting in a need for frequent recalibration. While recent attempts to create inherently robust decoders [20, 44] have demonstrated potential in mitigating this issue, they only postpone the inevitable need for recalibration.

In this work, we present an alternative approach called **CORP**: **C**ontinual **O**nline **R**ecalibration with **P**seudo-labels. CORP leverages the structure in language to enable self-recalibration of communication iBCIs without interrupting the user (i.e., plug-and-play). Specifically, CORP uses language models (LMs) to automatically correct communication iBCI text outputs, and continually retrains the decoder using these corrected outputs ("pseudo-labels"). LMs calculate probabilities for word sequences based on their likelihood of occurrence in the language, placing a strong prior on what the decoded text should look like that can help correct errors that would otherwise result in improbable sentences. Using an LM thus results in decoder outputs that contain fewer errors than the raw iBCI decoder outputs alone. CORP uses online stochastic gradient descent (SGD) to retrain the iBCI decoder to produce the pseudo-labels using the Connectionist Temporal Classification (CTC) [16] loss. We use various techniques such as a replay buffer and data augmentation to overcome the challenge of online SGD with very limited data. Note that CORP is agnostic to the communication task and can recalibrate any iBCI decoder that maps input neural signals to text outputs. CORP is particularly beneficial in scenarios where users rely on the iBCI system for continuous communication, and supervised data collection is either impractical or disruptive.

We built a prototype self-recalibrating handwriting iBCI system and assessed its long-term performance with one participant in a pilot clinical trial. Over a period of more than one year (403 days), our participant used the iBCI system monthly and wrote on average $58.3$ sentences per usage session. With CORP, the average accuracy was stable at $93.84\% \pm 2.28$ (mean $\pm$ standard deviation), significantly outperforming other recalibration methods. To our knowledge, our assessment represents the longest-running iBCI recalibration study to date, lasting 9 months longer than previous work [20]. CORP has the potential to remove a significant barrier to plug-and-play communication iBCIs. This study represents an initial step in applying machine learning (ML) methods that address nonstationary data distributions to neural network-based iBCI decoders. We believe that collaboration between the ML and iBCI community can lead to improved methods that specifically target the type of data distribution shifts seen for iBCIs. To that end, we released the data and code here[1] for future iBCI stability research.

---

[1] https://github.com/cffan/CORP

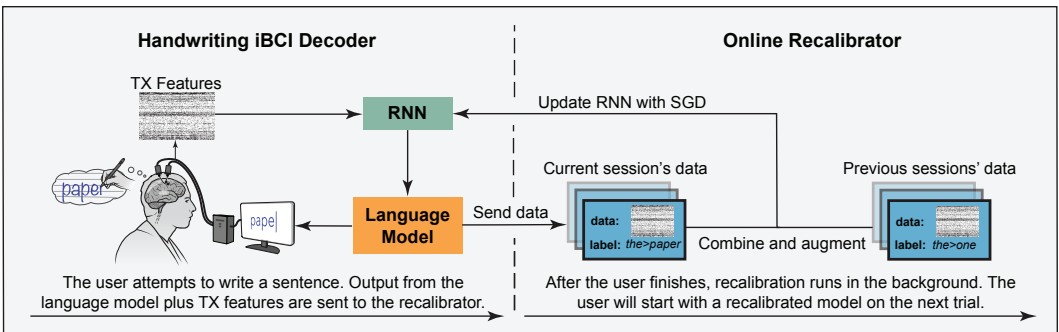

Figure 1: **Overview of the self-recalibrating handwriting iBCI system.** On the left is the handwriting iBCI decoder that decodes threshold-crossing (TX) neural features into text in real-time using an RNN and a language model (LM). On the right is the online recalibrator that continually updates the RNN decoder with pseudo-labels from the LM. The recalibrator mixes new data with previously collected data and uses stochastic gradient descent (SGD) to update the RNN.

## 2 Related work

Many unsupervised recalibration algorithms have been developed for iBCI decoders that attempt to transform neural data from new days to look like neural data collected on earlier days by aligning their distributions [10, 11, 27, 24]. These algorithms rely on the assumption that the neural signals can be mapped to a stable low-dimensional subspace which encodes a majority of task-related modulation [10, 3, 14, 45]. Unsupervised recalibration algorithms have achieved some success in cursor iBCIs, where a user controls a cursor on a computer screen via iBCI, but have not yet been demonstrated for iBCIs that decode more complex behaviors like handwriting or speech. Attempted handwriting or speech may require a larger number of neural dimensions to decode successfully compared to cursor iBCIs, and hence may be more difficult to align. We found that the Factor Analysis Stabilizer [10], a recently-proposed distribution-alignment method, did not appear to provide a benefit on our handwriting iBCI task. Further comparisons between CORP and these unsupervised distribution-alignment methods should be done in the future. Additionally, it is worth noting that CORP has the potential to be integrated with these distribution-alignment approaches, whereby the neural data is initially de-noised using the distribution-alignment strategies. This combination could potentially harness the strengths of both methods.

Self-training is a technique that involves using a pre-existing classifier to generate predictions ("pseudo-labels") on a large unlabeled dataset, followed by training a new model using these pseudo-labels. This approach has been widely employed in various machine learning applications [51, 18, 23]. Self-training as a means to recalibrate iBCIs has been investigated for cursor tasks [22, 49, 26]. Our work extends iBCI self-training to handwriting decoding, a richer task setting with higher dimensional outputs. We also leveraged LMs to provide robust priors, resulting in stable performance over a more extended time period. Self-training in speech recognition also utilizes LMs [23]; however, our approach is distinct due to its unique ability to handle continually shifting data distributions, while also operating in the data-limited regime of online iBCI control. LM-guided recalibration has also been explored for non-invasive BCI [6]. However, the inputs and decoding methods of non-invasive BCI differ significantly from those of iBCI.

Continual learning is an approach that enables models to learn from a large number of tasks sequentially, without forgetting knowledge obtained from the preceding tasks [17, 31]. This is particularly important in the context of our work, where the iBCI decoder must continually adapt to changes in the neural signals over time. To address the issue of catastrophic forgetting in continual learning, a variety of techniques have been developed [13, 36, 37]. In this work, we employ the replay buffer technique [37]. In future research, we plan to investigate additional approaches to further enhance the performance of our iBCI recalibration method.

## 3 Methods

In this section, we give an overview of the handwriting iBCI system and the Continual Online Recalibration with Pseudo-labels (CORP) framework (Figure 1).

## 3.1 Handwriting iBCI

Our handwriting iBCI system was developed based on the framework outlined in [47]. To train and evaluate the system, the user was instructed to copy sentences displayed on a computer screen by attempting to handwrite them letter by letter. Neural activity was recorded with two 96-channel silicon microelectrode arrays (Blackrock Neurotech). These microelectrodes were placed in the "hand knob" region of the dorsal motor cortex of our clinical trial participant (full details in Section 4).

Binned threshold-crossing counts were used as features for neural decoding, and were computed by counting the number of times the voltage time series on a given electrode crossed an amplitude threshold set at $-4.5$ times the standard deviation of the voltage signal. The iBCI decoder comprises a shared Gated Recurrent Unit (GRU) RNN [8] backbone and day-specific affine transform layers for each recording session. The neural features first undergo an affine transformation via the day-specific layer before being processed by the shared GRU backbone. Given the small amount of collected data, we used a 2-layer GRU with 512 hidden units to decode the neural features into a sequence of character probabilities, comprising 26 English letters and 5 punctuation marks (periods, commas, apostrophes, question marks and spaces).

A 3-gram language model (LM), along with a beam search algorithm [34], processed the RNN character probability output and translated it into words in real-time, which were displayed on the screen. The 3-gram LM was trained on the OpenWebText2 [15] corpus using Kaldi [34]. It uses a 130,000 word vocabulary taken from the CMU Pronouncing Dictionary [1]. After the user finished writing each sentence, we employed the large language model (LLM) GPT2-XL [35] to rescore the 3-gram LM's n-best outputs, and the top-scored result was displayed on the screen as the final decoded sentence.

The iBCI decoder was trained using a combination of data from the same participant in [47] and newly collected data. The training was done with an NVIDIA A100 GPU and took about 2 hours. We used the Connectionist Temporal Classification (CTC) loss [16] as opposed to the cross-entropy (CE) loss when training the GRU. The advantage of using the CTC loss is that it eliminates the need for labeling the exact timing of each character while maintaining performance on par with a CE-trained model.

## 3.2 Continual online recalibration with pseudo-labels

Formally, the iBCI recalibration problem can be defined as follows. Let $f : x \mapsto y$ be an iBCI decoder that maps neural activity $x$ into text outputs $y$. $x \in \mathbb{R}^{T \times C}$, where $T$ is the number of time bins and $C$ is the number of features. Suppose we are given a pretrained iBCI decoder $f_{\theta_0}$ with model parameters $\theta_0$. $f_{\theta_0}$ has been trained on a dataset $\mathcal{D} = (x_{i,t}, y_{i,t} | t < k)$. $x_{i,t}$ denotes the neural activity for trial $i$ on day $t$ and $y_{i,t}$ denotes the corresponding ground-truth text label. $\mathcal{D}$ is collected before day $k$. On a new day $k$, we want to decode a new neural signal $x_{j,k}$ into text. Due to nonstationarity, $x_{j,k}$ does not come from the same distribution as $x_{i,t} | t < k$, and therefore decoder $f_{\theta_0}$'s parameters require updating:

$$\theta_{j,k} = \underset{\theta}{\operatorname{argmin}} \mathcal{L}(f_\theta(x_{j,k}), y_{j,k}) \tag{1}$$

$\mathcal{L}$ is a distance function that measures the difference between decoded text $\hat{y}_{j,k}$ and ground-truth text $y_{j,k}$. We use Levenshtein distance for $\mathcal{L}$.

There are two major challenges to solve the above problem. The first arises from the fact that we do not know $y_{j,k}$, as we want the user to continuously use the iBCI system without stopping for supervised data collection. The second challenge is that at the beginning of day $k$, we only have a very small sample of $x_{j,k}$ (e.g. from a few seconds to a minute). Fine-tuning neural network models on small datasets can lead to overfitting and catastrophic forgetting [31, 36]. In the following sections, we describe how we address these two challenges.

### 3.2.1 Pseudo-labels

Although we do not know the ground-truth label for $x_{j,t}$, the decoded output $\hat{y}_{j,k}$ is mostly accurate. In [47], the author reported a 1.5% character error rate when evaluating a pretrained decoder on a new day that is less than a week away. However, when evaluating on a new day that was more than two weeks away, the accuracy dropped significantly. This observation suggests that using

LM-corrected outputs as pseudo-labels for continual recalibration could potentially maintain the decoder's accuracy over time, as long as recalibration is performed frequently enough. Formally, we replace the ground-truth label $y_{j,k}$ with the pseudo-label $\hat{y}_{j,k}$ in Eq. 1.

### 3.2.2 Continual online recalibration

On each new day $k$, the recalibration process is initiated immediately after the user completes writing the first sentence. This process runs continuously in the background, updating the decoder after every sentence. Such a continual online recalibration poses an optimization challenge, distinct from traditional deep neural network optimization, which typically assumes access to large, correctly labeled datasets. In our case, we only have a limited amount of data with pseudo-labels, and due to within-day nonstationarity, the distribution of $x_{j,k}$ may be also subject to change. We surveyed the literature on continual learning to identify methods that would enable us to optimize our GRU decoder using small amounts of pseudo-labeled data.

**Replay buffer** A replay buffer in continual learning is a fixed-size buffer, typically denoted by $M$, which stores data pairs $(x_{i,t}, y_{i,t})$ [37]. Its purpose is to balance the learning process by combining new data with previously collected data, mitigating issues such as data sparsity, overfitting, and catastrophic forgetting. We use a replay buffer to combine data collected prior to day $k$ and data from day $k$. $p$ percent of the replay buffer is reserved for day $k$'s data and the remaining space is allocated for data from previous days. The incorporation of new data alongside existing data addresses the data sparsity issue, preventing the model from overfitting to a limited new dataset. Furthermore, this approach safeguards the model against forgetting previously learned information.

**Data augmentation** Data augmentation is a technique commonly employed in machine learning to artificially expand the training dataset by generating new samples through the application of various transformations. This approach enhances the model's ability to generalize by exposing it to a wider range of data variations, leading to improved performance and robustness. To enrich the recalibration data, we also employ data augmentation techniques. Following [47], we introduce two types of artificial noise to the neural features. Firstly, we add white noise directly to the input feature vectors at each time step, providing the model with a more diverse set of inputs. Subsequently, we introduce artificial random offsets to the means of the neural features, to make the decoder model more robust to baseline firing rates drift [22, 43, 10]. This combination of augmentation techniques serves to enhance the model's overall performance and stability.

**Stopping criteria** Online recalibration requires balancing time, accuracy, and nonstationarity. The recalibration process should be swift, enabling users to access the updated model promptly, while accounting for the time needed for neural network training to improve model accuracy. To strike a balance, we need to determine an optimal point for stopping the recalibration process in order to minimize waiting time and maximize accuracy. Note that unlike typical machine learning settings with fixed validation sets, we lack a validation set for two reasons: the first is that we have a very limited amount of data; the second is that due to nonstationarity, performing well on data collected in the past does not guarantee good performance in future.

Our approach involves setting a loss threshold during recalibration and selecting an appropriate learning rate. During recalibration, a training loss is computed for every gradient update step taken on the training data. Since lower training loss values correlate with higher accuracy (assuming no substantial overfitting has occurred), we stop the recalibration once the loss falls below a certain threshold. However, loss thresholds that are too low may prolong recalibration and risk overfitting to imperfect pseudo-labels. Thus, we must find an optimal loss threshold considering time constraints and pseudo-label quality. In addition to the loss threshold, the learning rate also affects the recalibration accuracy and time. A trade-off exists between small and large learning rates, with large rates requiring fewer gradient update steps but potentially causing instability, and smaller rates offering stability at the cost of increased time. The optimal learning rates and loss thresholds depend on the problem. In the next section, we conduct an offline sweep to analyze the trade-offs. We find a loss threshold and learning rate combination that effectively balances time, accuracy, and nonstationarity.

## 4 Results

We assessed the effectiveness of CORP with our clinical trial participant (referred to as 'T5'), who is part of the BrainGate2 pilot clinical trial. T5 is a right-handed man who was 69 years old at the time

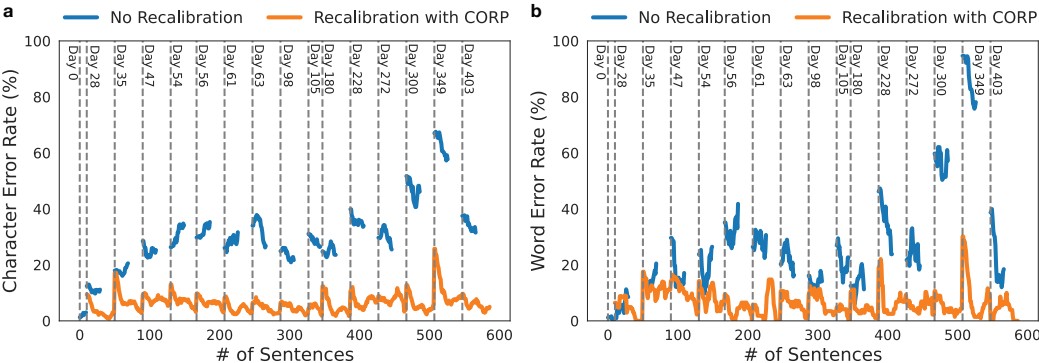

Figure 2: **An evaluation of CORP demonstrates high-performance over 403 days.** The character error rate (CER) and word error rate (WER) of each sentence across 15 evaluation sessions. Day-0 was to train and evaluate the seed model. Each subsequent day compared the no-recalibration blocks (blue) and the recalibration blocks (orange) where CORP was used. Details about each day can be found in the Supplement. (a) The CER of raw RNN-decoded outputs (pre-LM). (b) The WER of LM-decoded outputs.

of the study. He was diagnosed with C4 AIS-C spinal cord injury eleven years prior to this study. This study was conducted under an FDA investigational device exemption, with institutional review board approval (see details in Supplement).

Over a 403-day time period, T5 used our handwriting iBCI system in real-time each month to copy sentences displayed on the screen (15 evaluation sessions in total). Each evaluation session consisted of a side-by-side comparison of online handwriting performance with and without CORP recalibration. Every session was divided into four "blocks", with each block containing 10-20 sentences. T5 was instructed to read sentences displayed on a computer screen and attempt to write them. After finishing a sentence, T5 would verbally report this, and we would manually stop the decoder and start recalibration. T5's writing speed was $73.2 \pm 11.0$ characters per minute on average. All sentences were randomly sampled from the Brown corpus with no repetition [12].

The first block served as a warm-up for T5 to get reacquainted with the handwriting task after a hiatus. A fixed set of 10 sentences was used for the warm-up block. The second block employed a frozen seed model, trained on a combination of data from [47] and data collected prior to this evaluation (21 sessions in total). The third and fourth block featured recalibration using the same seed model but unfrozen, which was continually recalibrated using CORP. In a few sessions (details in the Supplement), the third and fourth block were run before the second block to mitigate the order effect.

**Plug-and-play long-term stability** CORP successfully maintained the seed model's high accuracy over a 403-day evaluation period without requiring any supervised data collection.

Figure 2a compares the character error rate (CER) of the RNN-decoded outputs (without an LM) between the no-recalibration blocks (original seed model) and the recalibration blocks (CORP-recalibrated model) for each evaluation day. On day-0, a seed model was trained and its CER was $3.22\%$, similar to what was reported in [47]. Starting on day-28, in the no-recalibration blocks the CER gradually increased and then fluctuated due to nonstationarity in the neural data. In contrast, the CORP recalibration blocks exhibited stable and low average CER of $6.35\% \pm 1.75$. During each evaluation day, the recalibration block's CER started higher than the day before but progressively decreased over the course of the day, demonstrating successful continual recalibration as the participant wrote more sentences. In some recalibration blocks (e.g. day-61), the CER initially decreased, increased, and then decreased again, revealing that nonstationarity can occur within a single day, and CORP can effectively address it. Notably, there were extended breaks between some sessions (e.g. day-105 and day-180) and CORP was still able to successfully recalibrate the decoder, showcasing its resilience and adaptability even in the face of long gaps in usage. Remarkably, the recalibration accuracy remains high and stable even after 405 days (6.18% CER). Figure 2b shows a similar comparison for the LM-decoded outputs, with performance measured in word error rate (WER) after the LM is applied. T5 could clearly perceive the performance difference between the no-recalibration blocks and the recalibration blocks, and expressed a preference for the system recalibrated with CORP.

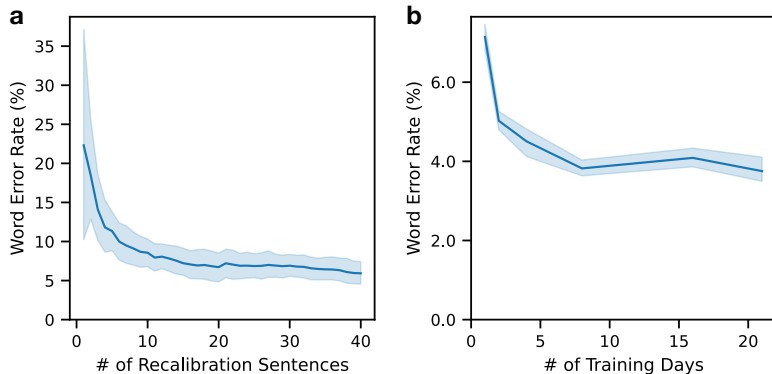

Figure 3: **Data needs for recalibration and seed model training.** Shaded region represents 95% confidence interval taken across 10 random seeds, computed via bootstrap resampling. (a) Word error rates continue to decrease as more sentences are used for recalibration, but only a handful of sentences are needed to regain the majority of the compromised accuracy. (b) Using more days of data to train the seed model enhances the recalibration accuracy, but performance does not appear to improve beyond 10 days.

CORP can achieve reasonable recalibration accuracy with minimal data. Figure 3a shows the recalibration WER as a function of the number of sentences used for recalibration (results were obtained from re-analyzing the data offline while using different amounts of training data). After approximately 10 sentences, CORP recovers most of the lost accuracy. Increasing the number of sentences can further improve recalibration accuracy (albeit at a reduced rate).

A high-quality seed model is important for maintaining consistent recalibration accuracy. Training a seed model on an extensive dataset allows the model to capture a wide range of variations in the underlying patterns, which in turn enhances its generalization capabilities. Figure 3b demonstrates the benefits of using more data for training. However, it also reveals a diminishing return after about 10 days' worth of data. This plateau likely occurs because the model has already learned a substantial amount from the available patterns and information within those 10 days, resulting in diminishing improvements as more data is added.

**Comparison of recalibration methods**    Table 1 compares the performance of CORP to other recalibration methods in online and offline settings. In the online setting, the average WER is measured using the recorded decoder outputs from all the online evaluation sessions. In the offline setting, we used the recorded neural activity from recalibration blocks from day-28 to day-228 to compare different recalibration methods. Offline recalibration was run with 10 random seeds to compute the confidence interval using bootstrapping.

Table 1: Word error rate (WER) comparison of different recalibration methods. The online results were average WER from the no-recalibration and recalibration blocks. The offline results were computed using the recalibration blocks from day-28 to day-228. CORP's offline performance was improved due to hyperparameter search. The 95% confidence interval was computed via bootstrap resampling across 10 random seeds.

| Method | Offline WER% [95% CI] | Online WER% |
|---|---|---|
| No Recalibration | 23.15 [-, -] | 26.51 |
| FA Stabilizer | 23.03 [22.64, 23.27] | – |
| CORP | **3.68** [3.21, 4.76] | **6.16** |
| CORP (Ground Truth) | 2.05 [2.00, 2.12] | – |

Offline analysis allowed us to compare CORP to an alternative approach: unsupervised stabilization of the distribution of neural data. We compared CORP to the Factor Analysis (FA) Stabilizer, an unsupervised recalibration method originally developed for cursor iBCI tasks. The FA Stabilizer requires a reference day to estimate the FA subspace, and aligns all other days' data to that reference day. We used day-0 as the reference day and aligned all other days' data to day-0. An FA seed

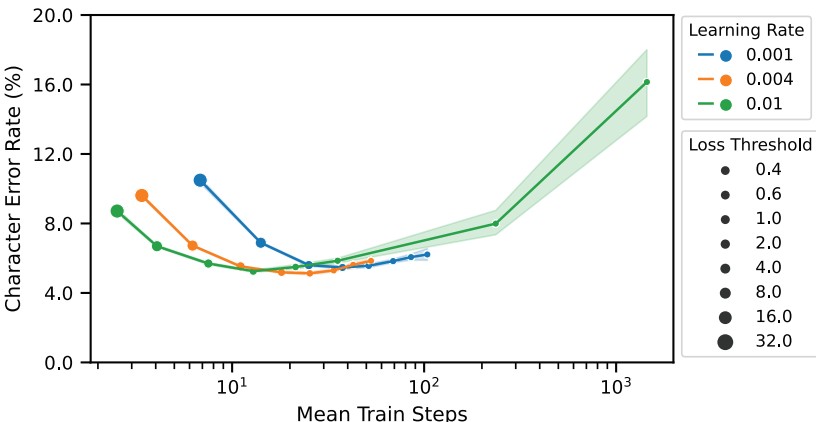

Figure 4: **Exploring the balance between recalibration accuracy and time requirements.** The relationship between recalibration accuracy, measured by character error rate (CER) on RNN-decoded outputs (as opposed to word error rate, which is noisier), and time required for recalibration, measured in terms of the number of gradient updating steps. Each color-coded line represents the varying loss thresholds at a constant learning rate. The size of the point represents the loss threshold. Shaded region represents 95% confidence interval taken across 10 random seeds, computed via bootstrap resampling.

model was trained on the aligned training data, frozen, and then evaluated on the aligned evaluation data. We used 96 latent dimensions for FA seed model training and recalibration as this was the best performing configuration. More details on the hyperparameters for the FA Stabilizer can be found in the Supplement. The results show that the FA Stabilizer was unsuccessful in the handwriting recalibration task, which stands in contrast to previous studies where it demonstrated success in a cursor iBCI task. This could potentially be because the FA Stabilizer assumes that a low dimensional latent structure is preserved across days and can be realigned using Procrustes alignment. Handwriting, however, has a higher intrinsic dimensionality than iBCI cursor control (see Supplement) and thus may be more challenging to realign. Nevertheless, many other promising distribution-alignment methods have been proposed [11, 27, 24], and future work should assess these methods and further probe their potential mechanisms of failure and improvement. In theory, distribution-alignment methods should be able to be combined with CORP for enhanced performance.

Note that the "No Recalibration WER" is different between the online and offline settings because, in the online setting, the no-recalibration blocks used different sentences from the recalibration blocks. When evaluating offline, we tested both methods on the same set of sentences. Additionally, the offline CORP has lower WER because we further optimized its hyper-parameters offline. We also compared CORP to continuous recalibration using the actual ground truth-labels, which sets a lower bound on the WER. With ground truth labels, we achieved a word error rate of 2%, showing that CORP is close to the lowest achievable error rate (1.6% difference), but with some room for improvement.

**Balancing recalibration accuracy and time**   CORP recalibration introduces minimal latency to the end-user's interaction with the system. On average, the recalibration procedure took approximately 9 seconds to complete (compared to an average of 36 seconds to write each sentence). During evaluation, the participant was requested to pause before writing the subsequent sentence until the recalibration was finished. However, in a real-world application, this recalibration could operate seamlessly in the background, unbeknownst to the user. The short run-time of CORP ensures that the recalibrated model is promptly available for use.

To achieve a balance between recalibration accuracy and time, we explored a range of loss thresholds and learning rates. Both loss threshold and learning rate together determine the recalibration time and accuracy, as shown in Figure 4. For any fixed learning rate, initially, as the loss threshold is reduced, the recalibration CER also decreases, but then increases as the model begins overfitting to imperfect pseudo-labels. Very small learning rates (e.g., learning rate = 0.001) not only take more time to recalibrate but also lead to sub-optimal recalibration CER. Conversely, large learning rates might achieve the optimum faster, but can cause instability when the loss threshold is small (e.g., the

train steps and CER increase dramatically for learning rate = 0.01 when the loss threshold is less than 1). Overall, CORP can work well with a wide range of learning rates and loss thresholds. This is advantageous for its practical use, as it eliminates the need for extensive hyperparameter tuning for each new user.

**Ablation on CORP** Table 2 presents the effects of various ablations on the accuracy of CORP's recalibration, highlighting the importance of certain key components.

Table 2: Ablation of key components in CORP, measured by character error rate (CER) on RNN-decoded outputs to make all ablations comparable to the "No Language Model". The 95% confidence interval was computed via bootstrap resampling across 10 random seeds.

| Ablation | CER% [95% CI] |
|---|---|
| CORP | 4.80 [4.68, 4.98] |
| No Data Augmentation | 5.87 [5.77, 6.01] |
| No Replay Buffer | 6.42 [6.33, 6.52] |
| No GRU Backbone Training | 7.00 [6.82, 7.19] |
| No Language Model | 19.22 [17.29, 22.18] |

The Language model (LM) makes the most significant contribution to recalibration accuracy. When the LM is not incorporated, as shown in the "No Language Model" ablation, the CER increases significantly when relying solely on RNN-decoded outputs as pseudo-labels. This underscores the LM's proficiency in correcting decoding errors induced by nonstationarities, and emphasizes its significance in the CORP framework.

The next ablation, "No GRU Backbone Training", freezes the shared GRU backbone of our decoder while only training the day-specific affine input layer. Since CER only increases by 2% if the backbone is frozen, this ablation results suggest that nonstationarities largely cause the neural representation to gradually rotate from the original space into a new one, aligning with previous findings [47]. Nonetheless, recalibrating both the backbone GRU and the day-specific layers yields improved recalibration accuracy, indicating that factors beyond mere rotation of the neural representation are in play.

Lastly, the ablations "No Replay Buffer" and "No Data Augmentation", which remove the use of the replay buffer and noise augmentation respectively, both result in a CER increase. Despite the increase being modest, these components may still be important during the initial stages of recalibration when data availability is limited.

**Limitations** One limitation of CORP lies in its dependence on the accuracy of the pseudo-labels. In Figure 5, we examine the impact of pseudo-label accuracy on recalibration accuracy. To simulate a varying pseudo-label accuracy, we randomly altered a specific percentage of letters in the ground truth sentences (replacing letters, inserting new letters, and deleting letters). As the CER of the pseudo-labels increased, the recalibrated RNN's CER also increased, following an almost linear relationship. The sublinear relationship can likely be attributed to the redundancy of letters in sentences. Interestingly, after auto-correction by an LM, the recalibration CER increased much more slowly than the pseudo-label CER. This again shows the importance of leveraging an LM for pseudo-label-based recalibration.

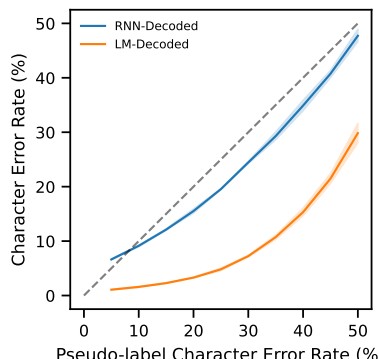

Figure 5: **Recalibration error rate increases sub-linearly with respect to the pseudo-label error rate.**

Considering these findings, we do not anticipate pseudo-label quality to be a major concern in practice. This is because future clinically viable iBCIs are expected to have a high decoding accuracy (which should be achievable with higher channel count devices, e.g. [28, 39, 32]). Users are also likely to utilize the iBCI frequently, resulting in small nonstationarities most of the time. Under

these conditions, we believe that the pseudo-labels will have high accuracy, allowing CORP to sustain the iBCI's accuracy indefinitely.

## 5 Discussion

iBCIs offer the potential to restore communication abilities lost due to neurological disease or injury. In this study, we presented CORP, a continual online recalibration method to address a nonstationarity of neural signals, which is a key challenge in iBCIs. CORP leverages language models to perform self-recalibration of any communication iBCI that maps neural activity to text. Through an extensive evaluation of a handwriting iBCI system, we demonstrated the efficacy of CORP in maintaining high decoding accuracy over an extended period of more than one year (403 days), achieving plug-and-play stability. This is the longest-running iBCI stability demonstration involving human participants. Our findings have important implications for the clinical translation of iBCIs. By maintaining decoding accuracy in the presence of nonstationarity, CORP can help to ensure reliable and stable iBCI performance, which is crucial for user acceptance and practical utility. Additionally, the reasonable time and computational resources associated with CORP facilitates the implementation of iBCIs in various clinical and home settings.

However, there are some limitations to the current study. The reliance on pseudo-label accuracy in CORP may pose challenges in cases where the iBCI system exhibits substantial nonstationarity or low decoding accuracy. Nonetheless, we expect that future iBCIs will continue to improve in performance as recording devices improve (e.g., [28, 39, 32]), mitigating this concern. Furthermore, the present study focused on a handwriting iBCI system with one participant, and the generalizability of our findings to other iBCI tasks and more participants remains to be established.

In future work, several avenues could be explored to further enhance the recalibration process. For instance, combining distribution-alignment approaches [10, 11, 24, 27] with CORP might harness the strengths of both methods. Various methods developed for addressing distribution shift in machine learning could also be investigated [41, 2, 5, 19, 25, 21, 38, 42]. It is also important to examine the performance of CORP in other iBCI applications, such as speech decoding[48], to gain valuable insights into its broader applicability and potential for further advancements in the iBCI field.

In conclusion, CORP represents a promising approach to address the challenges of nonstationarity in iBCI systems. By enabling effective online recalibration without the need for additional supervised data collection, CORP has the potential to tackle one of the remaining technical roadblocks for clinically translating high-performance communication iBCIs.

## 6 Acknowledgements

We thank participant T5 and his caregivers for their generously volunteered time and effort as part of the BrainGate2 pilot clinical trial; B. Davis, K. Tsou and S. Kosasih for administrative support. Support was provided by the Office of Research and Development, Rehabilitation R&D Service, Department of Veterans Affairs (nos. N2864C and A2295R), Wu Tsai Neurosciences Institute, Howard Hughes Medical Institute, Larry and Pamela Garlick, Simons Foundation Collaboration on the Global Brain and NIH-NIDCD U01DC017844, NIH-NIDCD R01DC014034, NIH-NIBIB R01EB028171.

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
