# Contents

# 1 Handwriting iBCI

## 1.1 RNN training details

This section lists the details for training the Gated Recurrent Unit [4] RNN which was used as the handwriting iBCI decoder.

**Training data** The RNN was trained on a combination of data from [15] (10 recording sessions) and newly collected data (11 recording sessions Table 2).

**Feature pre-processing** The recorded neural voltage data were converted into threshold-crossing (TX) features first by counting the number of times the voltage time series crossed an amplitude threshold set at $-4.5$ times the standard deviation of the voltage signal. TX features were then pre-processed by binning into 20ms time steps, "z-scoring" (subtracting the mean and then dividing by the standard deviation), and causally smoothed by convolving with a Gaussian kernel (sd = 40ms). Finally, the data was subsampled by a factor of 2.

**Data augmentation** TX features were augmented by adding two types of artificial noise. Firstly, random Gaussian white noise ($mean = 0\ std = 1.2$) was added to the feature vector at each time step. Subsequently, random constant offsets ($mean = 0\ std = 0.6$) were added to the means of the TX features.

$$x'_t = x_t + \epsilon_t + \phi \tag{1}$$

Here, $x'_t$ are the neural features with noise added, $x_t$ are the original neural features, $\epsilon_t$ is a white noise vector unique to each time step, and $\phi$ is a constant offset vector.

**Day-specific affine transform layer** The day-specific affine transform layer is defined as:

$$y = Ax + b \tag{2}$$

where $x \in \mathbb{R}^{c \times 1}$ is the input neural features and $c$ is the input dimension. $A \in \mathbb{R}^{c \times c}$ and $b \in \mathbb{R}^c$ are the parameters. Each session day has its own affine transform layer. The affine transform layers are trained together with the RNN. For a new session, a new affine transform is created and its weights are initialized with the previous session's. During online decoding, the input neural features are transformed by the affine layer first before being processed by the RNN.

41 **RNN training hyperparameters**    The hyperparameters for RNN training are listed in Table 1. The
42 training was done with one NVIDIA A100 GPU, taking about 2 hours.

Table 1: RNN training hyperparameters

| Description | Hyperparameter |
| --- | --- |
| Learning rate | 0.01 |
| Batch size | 48 |
| Number of training batches | 20000 |
| Number of hidden units in the GRU | 512 |
| Number of GRU layers | 2 |
| Dropout rate in the GRU | 0.4 |
| Optimizer | Adam |
| Learning rate decay schedule | Linear |
| L2 weight regularization | 1e-5 |
| Maximum gradient norm for clipping | 10 |

## 1.2    Language model training details

44 The 3-gram language model (LM) was trained using the SRILM [14] and then converted into a
45 weighted finite-state transducer (WFST) [10] with Kaldi [11].

46 The 3-gram LM was trained on the OpenWebText2 corpus [6] , which was pre-processed to include
47 only 26 English letters and 5 punctuation marks (periods, commas, apostrophes, question marks,
48 and spaces). It used a 130,000 word vocabulary taken from the CMU Pronouncing Dictionary [1].
49 Out-of-vocabulary words were mapped to a special <UNK> token. Witten-Bell discounting [17] was
50 used to improve the probability estimates of unseen or rare word combinations.

51 The 3-gram LM was then converted into a WFST, following the recipe in [9]. The WFST was
52 composed of three individual WFSTs:

$$T \circ L \circ G \tag{3}$$

53 Here, $\circ$ denotes composition. G is the grammar WFST that encodes legal sequences of words and
54 their probabilities based on the 3-gram LM. L is the lexicon WFST that encodes what letters are
55 contained in each word. T is the token WFST that maps a sequence of RNN output labels to a single
56 letter. In our case, T contains all 26 English letters, 5 punctuation marks, and the CTC blank symbol.

## 2    CORP online assessment details

58 This section lists details of the online assessment of CORP.

### 2.1    Study participant

60 Research sessions were conducted with volunteer participant T5 enrolled in the BrainGate2 pilot
61 clinical trial (ClinicalTrials.gov Identifier: NCT00912041). The trial is approved by the U.S. Food
62 and Drug Administration under an Investigational Device Exemption (Caution: Investigational device.
63 Limited by Federal law to investigational use) and the Institutional Review Boards of Stanford
64 University Medical Center (protocol #20804), Brown University (#0809992560), and Massachusetts
65 General Brigham(#2009000505).

66 Participant T5 is a right-handed man who was 69 years old at the time of the study. He was diagnosed
67 with C4 AIS-C spinal cord injury eleven years prior to this study. T5 is able to speak and move his
68 head, and has residual movement of his left bicep as well as trace movement in most muscle groups.
69 T5 gave informed consent for this research and associated publications.

### 2.2    Data collection sessions

71 All data collection sessions for this study are listed in Table 2. Sessions 1-11 were used for seed model
72 training. In each of those sessions, the participant copied sentences on a computer screen without
73 seeing feedback from the real-time decoder. Session 12-26 were recalibration assessment sessions.
74 Each assessment session consisted of a warmup block, a no-recalibration block, and two recalibration

75  blocks. In session 12-23 and 26, the blocks were ordered as warmup block, no-recalibration block,
76  and recalibration blocks. In session 24 and 25, the blocks were ordered as warup block, recalibration
77  blocks, and no-recalibration block.

## 2.3 Seed model training

79  Because [15] shared the same participant as ours, we combined its data to train our seed model.
80  On day 0 (session 11), we first collected 50 sentences and combined them with data from [15] and
81  session 1-10 to train the seed model. We then evaluated the seed model's online performance on
82  another 10 sentences to establish a baseline.

## 2.4 Online handwriting decoding

84  **Neural signal processing**   Neural signals were recorded from the microelectrode arrays using the
85  Neuroplex-E system (Blackrock Microsystems) and transmitted via a cable attached to a percutaneous
86  connector. Signals were analog filtered (4th order Butterworth with corners at 0.3 Hz to 7.5 kHz),
87  digitized at 30 kHz (250 nV resolution), and fed to custom software written in Simulink (Mathworks)
88  for digital filtering and feature extraction. Digital filtering began with a highpass filter (300 Hz
89  cutoff) that was applied non-causally to each electrode, using a 4 ms delay, in order to improve
90  spike detection [8]. After filtering, binned threshold crossing counts (20 ms bins) were computed by
91  counting the number of times the filtered voltage time series crossed an amplitude threshold set at
92  -4.5 times the standard deviation of the voltage signal.

93  **Data collection rig**   Digital signal processing and feature extraction was performed on a dedicated
94  computer using Simulink Real-Time. Extracted features were then sent to a separate computer
95  running Ubuntu for neural decoding and recording. Decoding and recording software was written in
96  Python using TensorFlow 2 and Redis. The Ubuntu computer also ran the experimental task software
97  that displayed cues to the participant on a computer monitor. The task software was implemented
98  using MATLAB and the Psychophysics Toolbox [3]). Finally, a third computer running Windows was
99  used to interface with the Neuroplex-E system and control the starting and stopping of experimental
100 tasks.

101 **Online handwriting decoding**   The online handwriting decoder consisted of an RNN and an
102 LM decoder. The RNN ran every 40ms to process a neural feature frame and output CTC label
103 probabilities. The LM decoder took the RNN probability output and ran beam search on the WFST
104 decoding graph. We used the beam search implementation in WeNet [18]. Following [12], a constant
105 penalty was added to the CTC blank label probability.

106 After the real-time decoding was done, we ran a second-pass rescoring using GPT2-XL on the n-best
107 outputs from the LM decoder:

$$score(s) = \alpha * log(P_{RNN}(s)) + \beta * log(P_{ngram}(s)) + (1 - \beta) * log(P_{gpt}(s)) \tag{4}$$

108 Here $P_{RNN}(s)$ is the CTC label sequence probability given by the RNN for sentence $s$. $P_{ngram}$ is
109 sentence $s$'s probability under the 3-gram LM. $\alpha$ is the scaling factor on the RNN's log probabilities.
110 $\beta$ is the interpolation weights between the 3-gram LM and GPT2-XL.

111 All hyperparameters are listed in Table 3.

112 **Rolling z-scoring**   During online decoding, we used a rolling estimate of the mean and standard
113 deviation of each feature to perform z-scoring. This helps account for neural nonstationarities that
114 accrue across time.

115 For the first sentence of a new block, we used the previous block's mean and standard deviation. For
116 each subsequent sentence, we used up to 10 sentences preceding it to compute the mean and standard
117 deviation.

## 2.5 Online recalibration

119 All hyperparameters for online recalibration are listed in Table 4. During the online assessment, we
120 used a relatively large loss threshold and set a minimum number of gradient update steps to ensure
121 good recalibration accuracy. However, in later offline analysis, we found this strategy to be less
122 optimal compared to using a smaller loss threshold without the minimum steps.

Table 2: Data Collection Sessions

| Session Number | Date | Description | Data |
|---|---|---|---|
| 1 | 2022.05.18 | Seed model data collection session | 50 sentences |
| 2 | 2022.05.23 | Seed model data collection session | 80 sentences |
| 3 | 2022.05.25 | Seed model data collection session | 60 sentences |
| 4 | 2022.06.01 | Seed model data collection session | 80 sentences |
| 5 | 2022.06.03 | Seed model data collection session | 80 sentences |
| 6 | 2022.06.06 | Seed model data collection session | 90 sentences |
| 7 | 2022.06.08 | Seed model data collection session | 50 sentences |
| 8 | 2022.06.13 | Seed model data collection session | 80 sentences |
| 9 | 2022.06.15 | Seed model data collection session | 60 sentences |
| 10 | 2022.06.22 | Seed model data collection session | 80 sentences |
| 11 | 2022.09.01 | Seed model data collection session | 60 sentences |
| 12 | 2022.09.29 | Recalibration assessment session | 10 warmup sentences
20 no-recalibration sentences
40 recalibration sentences |
| 13 | 2022.10.06 | Recalibration assessment session | 10 warmup sentences
20 no-recalibration sentences
40 recalibration sentences |
| 14 | 2022.10.18 | Recalibration assessment session | 10 warmup sentences
20 no-recalibration sentences
40 recalibration sentences |
| 15 | 2022.10.25 | Recalibration assessment session | 10 warmup sentences
20 no-recalibration sentences
37 recalibration sentences |
| 16 | 2022.10.27 | Recalibration assessment session | 5 warmup sentences
20 no-recalibration sentences
40 recalibration sentences |
| 17 | 2022.11.01 | Recalibration assessment session | 10 warmup sentences
20 no-recalibration sentences
40 recalibration sentences |
| 18 | 2022.11.03 | Recalibration assessment session | 10 warmup sentences
20 no-recalibration sentences
40 recalibration sentences |
| 19 | 2022.12.08 | Recalibration assessment session | 10 warmup sentences
20 no-recalibration sentences
40 recalibration sentences |
| 20 | 2022.12.15 | Recalibration assessment session | 10 warmup sentences
19 no-recalibration sentences
20 recalibration sentences |
| 21 | 2023.02.28 | Recalibration assessment session | 6 warmup sentences
20 no-recalibration sentences
40 recalibration sentences |
| 22 | 2023.04.17 | Recalibration assessment session | 10 warmup sentences
20 no-recalibration sentences
40 recalibration sentences |
| 23 | 2023.05.31 | Recalibration assessment session | 10 warmup sentences
20 no-recalibration sentences
40 recalibration sentences |
| 24 | 2023.06.28 | Recalibration assessment session | 10 warmup sentences
40 recalibration sentences
20 no-recalibration sentences |
| 25 | 2023.08.16 | Recalibration assessment session | 10 warmup sentences
40 recalibration sentences
20 no-recalibration sentences |
| 26 | 2023.10.09 | Recalibration assessment session | 10 warmup sentences
20 no-recalibration sentences
40 recalibration sentences |

Table 3: Beam search hyperparameters

| Description | Hyperparameter |
|---|---|
| Beam search min active states | 200 |
| Beam search max active states | 7000 |
| Beam size | 17 |
| Acoustic scale | 0.8 |
| $\alpha$ | 0.5 |
| $\beta$ | 0 |
| Number of n-best outputs | 10 |
| Penalty applied on blank labels | log(11) |

Table 4: Online Recalibration hyperparameters

| Description | Hyperparameter |
|---|---|
| Min number of gradient update steps | 32 |
| Max number of gradient update steps | 200 |
| Loss threshold | 20 |
| Learning rate | 0.004 |
| Percentage of new data in the replay buffer | 0.6 |
| Batch size | 64 |
| Optimizer | Adam |

## 3   Offline analyses details

This section lists details of the offline analyses. All offline analyses used recalibration blocks from session 12-22. Session 22-26 were collected during paper review and thus not used for offline analyses.

### 3.1   Factor Analysis Stabilizer

We applied the Factor Analysis (FA) Stabilizer to the handwriting iBCI data as follows.

#### 3.1.1   FA Stabilizer seed model training

The FA Stabilizer assumes that neural activity tends to lie within a stable low-dimensional space, and that nonstationarities are largely caused by the rotation of this latent space. It uses Factor Analysis [2] to identify the latent low-dimensional space:

$$z_t \sim \mathcal{N}(0, \mathbf{I}) \tag{5}$$
$$x_t | z_t \sim \mathcal{N}(\mathbf{\Lambda} z_t + \mu, \mathbf{\Psi}) \tag{6}$$

$x_t \in \mathbb{R}^c$ is the neural activity (threshold-crossing counts on $c$ electrodes at time step $t$). $z_t \in \mathbb{R}^d$ is the low-dimensional latent representation of the neural activity. $\mathbf{\Lambda} \in \mathbb{R}^{c \times d}$ is the loading matrix that linearly transforms the neural activity into the latent space. $\mu \in \mathbb{R}^c$ is the mean mean spike counts for each electrode. $\mathbf{\Psi} \in \mathbb{R}^{c \times c}$ is a diagonal matrix that describes the variability that is independent for each electrode.

We picked session-11 as the reference day to estimate the loading matrix $\mathbf{\Lambda}_1$. For a new session, we first estimated its loading matrix $\mathbf{\Lambda}_2$. We then used the Procrustes analysis [13] to align those two latent spaces:

$$\hat{O} = \operatorname*{argmin}_{O:OO^T=I} \| \mathbf{\Lambda}_1 - \mathbf{\Lambda}_2 O^T \|_F^2 \tag{7}$$

$\hat{O} \in \mathbb{R}^{d \times d}$ is an orthogonal matrix. After $\hat{O}$ is identified, the new session's latent space can be aligned to the reference day's by $\mathbf{\Lambda}_2 O^T$.

Additionally, following the algorithm in [5], electrodes with large changes between sessions were iteratively removed. The iterative channel elimination algorithm uses two parameters: $B$ and $T$. $B$ determines the number of electrodes to be retained, while $T$ sets a threshold for the L2 norm of the load matrix's row. If the norm falls below this threshold, the corresponding row will be eliminated.

147 We ran a grid search on $B$ and $T$ and found that a wide range of $B$ ($B \geq 110$) and $T$ ($0.01 \leq T \leq 0.1$)
148 all worked well for the handwriting iBCI task. We used $B = 160\ T = 0.1$ in all our experiments.

149 The seed model for the FA Stabilizer was trained using all sessions leading up to and including the
150 reference session. After training, the seed model was frozen for all the recalibration evaluations.

### 3.1.2   Recalibration with FA Stabilizer

152 We evaluated the FA Stabilizer on all recorded online recalibration blocks.

153 For each evaluation day, we first used the no-recalibration block's data (20 sentences) to estimate
154 the initial alignment between the evaluation session and the reference session. Then for each new
155 sentence in the recalibration block, we pushed it into a sliding buffer of size 20 and used the data in
156 the buffer to estimate a new alignment. The aligned neural data for that sentence was then decoded
157 with the FA Stabilizer seed model.

### 3.1.3   FA dimensionality analysis

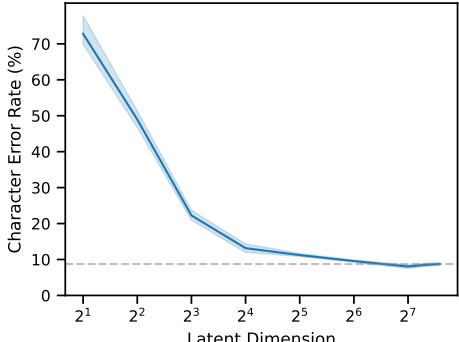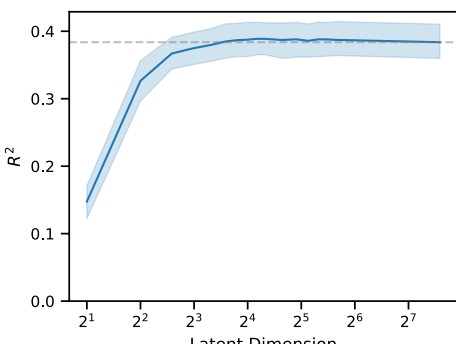

Figure 1: **Effect of FA dimensionality on handwriting and cursor iBCIs** (Left) Applying FA with
varying dimensionality (2-160) to a single handwriting session. The dashed line shows decoding
character error rate (CER) on the original data (with no FA applied). Approximately 100 dimensions
yield accuracy close to that of the original data. Increasing the dimensionality beyond 100 decreases
CER slightly, indicating that FA may help denoise the data. (Right) Applying FA with varying
dimensionality to the cursor data. Performance is measured in $R^2$ (higher values are preferable), with
the optimal dimensionality found to be around 10.

159 To find the optimal dimensionality for using the FA Stabilizer on the handwriting iBCI task, we
160 trained various FA Stabilizer seed models while sweeping the number of dimensions in the FA. The
161 seed models were trained on a single session (the reference day) without Procrustes alignment.

162 For the cursor FA dimensionality analysis, we applied FA with varying dimensionality to the cursor
163 iBCI data from [16]. Specifically, we picked sessions with more than two blocks of cursor control
164 data. The raw neural recordings were pre-processed using the the pipeline as the handwriting iBCI
165 data (no subsampling). We then did a 50-50 train and test split. For each session, we first trained 10
166 iterations of FA at each dimensionality. We selected the model with the highest log likelihood for the
167 training data. We then built a simple Ridge regression from the neural activity in FA subspace to the
168 instantaneous cursor-to-target vector. We swept the regularization strength (1e1, 1e3, 1e5, 1e7, 1e9)
169 on the held-out test sets.

170 We found that unlike the cursor iBCI task, where ~10 dimensions are enough to saturate the task
171 performance, the handwriting iBCI task needs ~100 dimensions (Figure 1). Identifying the reasons
172 why handwriting decoding benefits from including more neural dimensions is an interesting direction
173 for future research.

### 3.2   Additional offline analyses

175 **Artificial noises augmentation**   We added two kinds of artificial noise to the recalibration data. We
176 analyzed the effects on recalibration accuracy when varying the magnitude of each type of noise in

Figure 2. The results showed that while adding white noise improved performance, adding random offsets to the feature means did not. This could be because, during the recalibration sessions the feature means changed slowly, and online z-scoring already removed the effects of this kind of slow mean change.

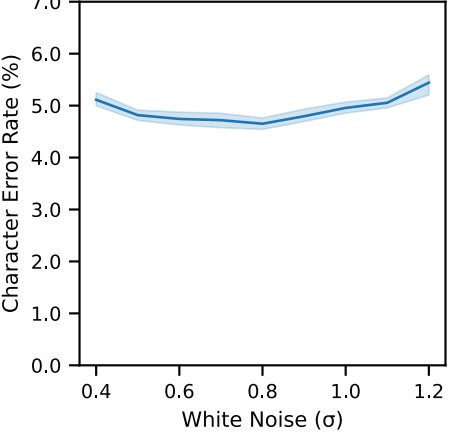 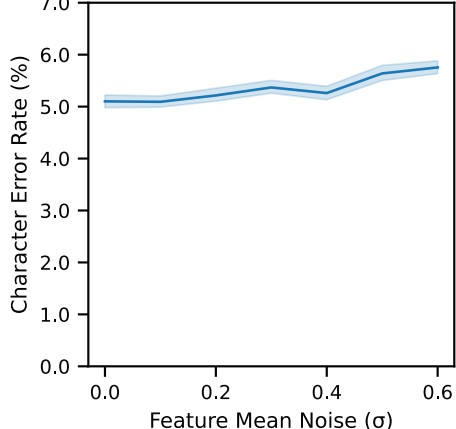

Figure 2: **Effects of artificial noise augmentation on recalibration accuracy.** (Left) Adding a small amount of white noise to the recalibration data improved recalibration accuracy. (Right) Adding random constant offsets to the feature means did not improve recalibration accuracy. This is likely due to feature means changing slowly during recalibration sessions, in a way that was successfully accounted for already by online z-scoring.

**Percentage of new data included in the replay buffer** The replay buffer has a parameter $p$ that controls the percentage of new data. During online evaluation, we loaded all past sessions' data into the replay buffer, and randomly sampled $batch\_size \times p\%$ sentences of new data, and $batch\_size \times p\%$ of old data. In Figure 3, we analyzed the effect of $p$ on recalibration accuracy. A wide range of parameters (10% - 70%) all worked well, indicating that only a small amount of past data is needed to prevent catastrophic forgetting.

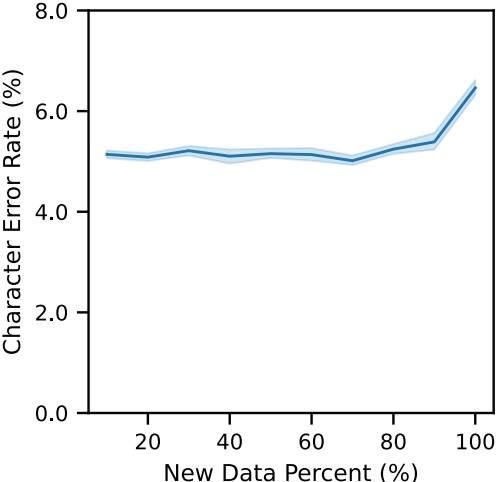

Figure 3: **Effect of the percentage of new data included in the replay buffer on recalibration accuracy.** Only a small percentage of old data is needed to keep the model from catastrophically forgetting.

**3-gram vs. GPT2-XL** CORP used a 3-gram LM for the first pass decoding to generate 10 decoding hypotheses, then used GPT2-XL to rescore these hypotheses. The final LM-decoded result was

used as a psuedo-label for recalibration. Figure 4 shows how pseudo-labels generated by different LMs affect the recalibration accuracy. It shows that the advantage of using GPT2-XL to rescore the 3-gram hypotheses is only marginal. This can be attributed to two factors. First, the 3-gram decoding accuracy is already close to the ground truth accuracy, leaving little room for improvement. Second, GPT2-XL is not as powerful as more recent large language models (LLMs) [7]. A comparison with more recent LLMs remains a topic for future exploration.

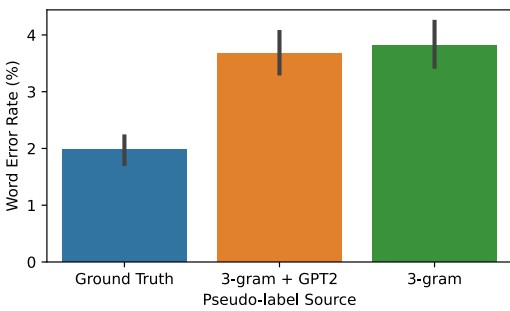

Figure 4: **Influence of different LMs on the recalibration accuracy when using CORP**. Using GPT2 in addition to the 3-gram LM improves recalibration accuracy only slightly. Both are close to the performance ceiling (using ground truth labels for recalibration).