# OpenReview forum: "Plug-and-Play Stability for Intracortical Brain-Computer Interfaces: A One-Year Demonstration of Seamless Brain-to-Text Communication"
_NeurIPS.cc/2023/Conference — NeurIPS 2023 spotlight_

### Official Review · Reviewer_QkLb · 2023-07-06

**Soundness:** 3 good
**Presentation:** 3 good
**Contribution:** 3 good
**Rating:** 7
**Confidence:** 3

**Summary:**

This paper introduces a method called Continual Online Recalibration with Pseudo-labels (CORP) that allows for self-recalibration of intracortical brain-computer interfaces (iBCIs) without the need for user interruption. iBCIs are used to restore communication abilities in individuals with neurological disorders like ALS but require frequent recalibration due to changes in neural recordings over time. The proposed method utilizes large language models (LMs) to automatically correct errors in iBCI outputs and uses these corrected outputs as "pseudo-labels" to update the iBCI decoder in real time. The CORP framework was evaluated over an 8-month period with a single participant in a clinical trial. The results showed a stable decoding accuracy of 93.71% in an online handwriting task, outperforming other baseline methods. This study demonstrates the longest-running iBCI stability with a human participant and presents a plug-and-play, high-performance communication iBCI that addresses a significant challenge in the clinical application of iBCIs.

**Strengths:**

1. originality: This work presents an online recalibration method for iBCI which uses LLM to perform re-calibration for communication iBCI.
2. quality and clarity: This paper provides detailed experimental validation of the effectiveness of the proposed method. The figures and text are clear, easy to understand, and flow smoothly, ensuring a clear expression of ideas.
3. significance: The CORP method proposed in this paper achieves higher accuracy and stability in brain-to-text communication through intracortical brain-computer interfaces (iBCIs) than other recalibration methods. It has been validated on the longest-running iBCI involving human participants, proving its effectiveness and potential.

**Weaknesses:**

1. The method presented in this paper has only been validated on a single subject, which may raise concerns about the limited sample size and lack of generalizability.
2. The stopping criteria mentioned in the article are based on whether the loss falls below a certain threshold to determine the termination of training. This criterion may not be sufficiently objective. It could be considered to explore alternative quantitative metrics for evaluating the training performance of the model.

**Questions:**

If a participant does not use the iBCI for an extended period without recalibrating the RNN model, the initial Word Error Rate (WER) and Character Error Rate (CER) can be very high. In such cases, using pseudo-labels generated with the help of LLMs may potentially mislead the training of the RNN model. Has this scenario been tested or are there any alternative approaches?

**Limitations:**

As mentioned by the authors in the paper, the main limitation of CORP lies in the pseudo-labels generated by the language model. If the error rate of the pseudo-labels is high, it directly affects the Character Error Rate (CER) of the recalibrated RNN model. Therefore, it is crucial to select more powerful language models to minimize errors in the pseudo-labels.

---

> ### Author Rebuttal · Authors · 2023-08-07
>
> > **Weakness 1**
>
> We acknowledge the limitation of low sample size. In the future, we plan to address this limitation by collaborating with other BCI labs to test our method on a broader range of subjects. We believe that such collaborations will provide valuable insights and help us refine our approach. Additionally, we plan to publish the code associated with our method. This will allow other researchers to test our method on their data, further contributing to the validation and generalizability of our findings.
>
> > **Weakness 2**
>
> We agree that the stopping criteria based on whether the loss falls below a certain threshold may not be sufficiently objective. We will consider alternative quantitative metrics for evaluating the training performance of the model in future work.
>
> > **Question 1**
>
> In our paper, we have attempted to address this issue in the limitation section by simulating varying pseudo-label accuracy. We acknowledge that our method could fail if the initial error rate is too high, typically indicating that the neural data has changed significantly. In such cases, supervised recalibration could be used to rescue the system. However, this is an area where more research is needed. Understanding the nature of the nonstationarity in neural data is a complex challenge, and further investigation is required to develop methods that can automatically address this problem more effectively.
>
> > **Limitations**
>
> Thank you for the suggestion of using more powerful language models. Recent advancements in large language models (LLMs) offer promising avenues for improving the accuracy of pseudo-labels which we plan to explore in future work.

---

### Official Review · Reviewer_hfNe · 2023-07-07

**Soundness:** 4 excellent
**Presentation:** 4 excellent
**Contribution:** 4 excellent
**Rating:** 8
**Confidence:** 4

**Summary:**

The paper proposes a self-recalibrating brain-computer interface system where inputs come from implanted electrodes in patient's motor cortex and the outputs characters that are patient images to hand-write. The output is then corrected by a language model to match to the most probable desired sentence.

The recalibration in needed because in the real biological system there are many factors that change over time and the input data distribution drifts because of that. While it is possible to recalibrate the system by running a dedicated session, this is not optimal is it requires downtime, participation of a technician and ~daily mental effort from the patient.

This work proposes a smart trick to used LLM-corrected word predictions as new labels for continuous retraining: the input signal distribution drift is gradual on the next day, while the drift has already occurred, the language model can still correctly recover what the intended word were. And these corrected words can be used as new labels to retrain the system a bit and this cancel the drift. If performed often the system will never drift too far and will always be able to self-correct.

The experimental results from one test subject with implanted microelectrode array confirm that the error rate of the system stays stable thanks to continuous recalibration and it is shows that it would degrade without the proposed mechanism.

**Strengths:**

The paper is very clearly written and well-structured, to the extent that I find myself in trouble performing my role as a reviewer's and ask questions, because I had to delete most most of them as I progressed through the pages :)

The application is, of course, amazing and the fact that this work was tested on a human subject with implanted electrodes makes it a unique contribution to the field of ML applications to BCIs.

**Weaknesses:**

I could not come up with any. I think paper is a very clear account of what was done and presents sufficient support for the claims.

Perhaps more test subjects would be beneficial, but even with one subject the results stand.

**Questions:**

Fig 1: What does the arrow from "Language model" to "Current session's data" mean? Is it somehow part of LM's training to predict session data? Or does it provide labels and the data in "Tx Fetures" is the same as in "Current session's data"?

126: How does the system know that the subject has finished thinking a sentence? Is there a special "stop"-though and if so - how reliably it can be detected?

Q1: What was X's subjective feedback on the performance of the system after a few months? While the numbers clearly show that the system was more accurate, what it also noticeable from the user's perspective?

Q2: How are spaces handles? I would image there is no clear motor imagery to think about to imagine making a space, how does the patient do that?

361: Perhaps this should go to Limitations?

**Limitations:**

I would be curious to know about the natural limits of speed of writing with this approach. After all having a direct brain connection would seem like an opportunity to forgo "clunky" written language and make a more direct transmission possible. Just curious what are the authors' thought on feasibility of this and what could be the way to do it.

---

> ### Author Rebuttal · Authors · 2023-08-07
>
> > **Question 1**
>
> The arrow from "Language model" to "Current session's data" represents the generation of pseudo-labels by the language model. We will clarify this in the revised manuscript.
>
>
> > **Question 2**
>
> In this study, the participant would indicate to us when they had finished, and we would manually stop the decoder (since this particular participant can still speak, although he is paralyzed from the neck down). However, we believe the end of the sentence could automatically be detected without much difficulty, and this would be important for a locked-in user (the main target population of such a device)  One potential method is to use the output of the RNN itself. The RNN we used in this study is capable of outputting a special "blank" state when the participant is not writing. If the RNN outputs this blank state for more than a certain threshold of time (t seconds), we can infer that the participant has finished writing.
>
>
> > **Question 3**
>
> Yes, our participant X provided positive feedback about the self-recalibration system. He reported that he could clearly perceive the performance difference between the no-recalibration blocks and the recalibration blocks, and he expressed a preference for the recalibration system. We will include X's subjective feedback in the revised manuscript.
>
> > **Question 4**
>
> A user would write‘>’ to indicate space, similar to [46].
>
> > **Question 5**
>
> We will mention the one subject limitation in the Limitations in the revised manuscript.
>
> > **Limitations**
>
> In our study, participant X achieved a writing speed of 69.5 ± 8.6 characters per minute. In a previous study [46], the authors reported that their participant was able to write as fast as 90 characters per minute. For a comprehensive review of different BCIs' communication speeds, refer to this blog post (https://www.paradromics.com/blog-post/enabling-connection-ii-bci-for-assistive-communication). As for the possibility of a more direct transmission between the brain and a computer, bypassing the need for written language, this is indeed a fascinating topic. However, achieving this goal would require significant advancements in our understanding of the brain and how it encodes information. Right now, intracortical BCIs focus on motor areas of the brain that represent the intention to move. Therefore, speeds are tied to motor production, for example how long it takes to write a letter or speak a sound. In the future, it may be possible to record from different areas of the brain that may bypass the need to formulate a motor intent, if the neural code can be understood.

---

> > ### Comment · Reviewer_hfNe · 2023-08-20
> >
> > Thank you for the clarifications!

---

### Official Review · Reviewer_8J1g · 2023-07-07

**Soundness:** 2 fair
**Presentation:** 3 good
**Contribution:** 3 good
**Rating:** 5
**Confidence:** 3

**Summary:**

There is a vastly growing literature reconstructing continuous language from non-invasive brain recordings using popular deep-learning models. These papers typically use invasive recordings from surgically implanted electrodes, while decoders that use non-invasive recordings can only identify stimuli from among a small set of letters, words, or phrases.

This paper contributes to that literature by introducing a new approach, CORP, a continual online recalibration method for intracranial Brain-Computer Interface (iBCI) that maps neural activity to text. Specifically, the authors use a language model (RNN) to automatically correct errors in iBCI outputs by continually updating the iBCI decoder with an online learning method. The experimental results revealed that the proposed framework achieved a stable decoding accuracy of 93.71% in an online handwriting iBCI task, significantly outperforming other baseline methods.

**Strengths:**

The paper contains the following key contributions:

* The novelty of this work: Different from previous continuous language reconstruction methods from non-invasive brain recordings, the authors build a framework to leverage the structure in language to enable self-recalibration of communication iBCIs without interrupting the user.
* The proposed framework is agnostic to the communication task and can recalibrate any iBCI decoder that maps input neural signals to text output.

Originality:

* The prototype of a self-recalibrating handwriting iBCI system and assessed its performance over a period of time is interesting.

**Weaknesses:**

Weaknesses:

* Although the paper's main idea is quite interesting, however, the experiment was conducted on one participant in a pilot clinical trial. Hence, the proposed method needs to be tested on more subjects.

* Authors have not discussed recent brain decoding works:

    [1] Semantic reconstruction of continuous language from non-invasive brain recordings, Jerry Tang, Amanda LeBel, Shailee Jain, Alexander G. Huth

    [2] Decoding speech from non-invasive brain recordings, Alexandre DÃ©fossez, Charlotte Caucheteux, JÃ©rÃ©my Rapin, Ori Kabeli, and Jean-RÃ©mi King

* In the above works, the authors reconstruct the continuous language from non-invasive brain recordings (fMRI and MEG) with better perplexity.

* Did the authors compare their methods on any existing datasets?

* Since the RNN language model is limited in handling long-term memory information and vanishing gradient problems, did the authors try with recent pretrained Transformer language models? We may expect better accuracy since the Transformer model was pretrained on larger corpora. Moreover, we may expect that a replay buffer is not required in the case of Transformer models.

Quality: The paper supports its claims with few details. Specifically, the methodology and experimental details are limited.

Clarity: The paper needs a lot of improvement with clear methodology details and motivation behind using the RNN model. The information provided in the submission needs to be more comprehensive to reproduce the results.

**Questions:**

* Did the authors interpret the states of RNN in a continual online learning setup? Is there any trend across days? Which information is overwriting more in the language model?

**Limitations:**

The authors have discussed several limitations and made future directions for the research community.

---

> ### Author Rebuttal · Authors · 2023-08-07
>
> We appreciate your time and effort in reviewing our manuscript. We would like to clarify that our work differs significantly from the recent non-invasive language decoding works. We'll make this difference clear below, and would like you to consider re-evaluate our work.
>
> > **Weakness 1**
>
> We acknowledge the limitation of the low sample size and understand the need for more extensive testing across additional users. Conducting studies with intracortical brain-computer interfaces (iBCIs) presents unique challenges, particularly in recruiting participants, compared to non-invasive BCIs. The nature of iBCIs, which involve invasive procedures, inherently limits the number of participants we can involve in our studies. However, we are committed to addressing it in the future work by collaborating with other iBCI research labs and testing our method on more participants.
>
> > **Weakness 2&3**
>
> We appreciate your suggestion to compare our method with recent brain decoding works. However, it's important to note that the context and objectives of our work differ significantly from those of the studies you mentioned.
>
> Our primary aim is to develop clinically viable BCI devices that can restore communication for individuals with paralysis. Two crucial requirements guide our work: the user must be able to freely express themselves by typing whatever message they desire, and the system must maintain high accuracy. In our referenced work [46] where a handwriting iBCI was first demonstrated, the authors used an RNN to decode brain signals into text with an accuracy exceeding 95% (measured as character error rate). Their method is general enough to express any English sentence. This paper extends [46] to address the issue of non-stationary neural recordings. We used n-gram language models and large language models to automatically correct the RNN decoder’s output, which is then used for online recalibration of the RNN. Our results indicated that with CORP, the RNN decoder’s performance can be maintained around 95% for a long period of time without burdening the user to collect new calibration data.
>
> The two papers you mentioned aim to solve a different problem. They attempt to decode text from brain signals (recorded using fMRI or MEG) while the participant is listening to a speech. In a clinical setting, it's unclear how this approach could help restore communication, which would require decoding text when a participant is speaking. Moreover, their decoding method can only achieve better than chance accuracy. For instance, in [1] Table 1, the word error rate (WER) is > 90%, meaning that 9 out of 10 words decoded are incorrect. While performance might be better when assessed with semantic metrics, our goal is to enable the user to express the exact text of their intended message.
>
> In summary, while we appreciate the suggestion, a direct comparison between our work and the two mentioned papers is not entirely appropriate due to the significant differences in context, objectives, and methodologies. We hope this clarifies our position and thank you for your insightful feedback.
>
> > **Weakness 4**
>
> In the field of intracortical BCIs (iBCIs), the only datasets available are those that study the long-term stability of cursor decoding. Unfortunately, there are  currently no existing datasets specifically for handwriting iBCIs. Data collection for iBCIs is a challenging task, due to the very limited number of clinical trial participants. One of the significant contributions of our work is the 8-month long dataset we collected, which will be published alongside this paper. We believe that this dataset will be a valuable resource for the iBCI community, enabling researchers to explore more methods for addressing the issue of non-stationarity.
>
> > **Weakness 5**
>
> We appreciate your question regarding the use of Transformer models in place of RNNs. We would like to clarify that in this work no RNN language model was used. We used an RNN only to decode brain signals into character probabilities. A language model (3-gram LM + a transformer-based LLM) was then used to transform these probabilities into a string of words. The replay buffer was used to store recent data for the purpose of online recalibration of the RNN decoder. This is necessary regardless of the type of model used.
>
> We hope this clarifies the methodology of our work.
>
> > **Question 1**
>
> We appreciate your suggestion, but our RNN is not a language model. Please refer to the answer above. If you have more questions regarding the RNN decoder, we’d be happy to answer.

---

> > ### Comment · Reviewer_8J1g · 2023-08-15
> > **Thanks for the rebuttal**
> >
> > Dear authors,
> >
> > Thanks for the rebuttal.
> >
> > By considering authors' feedback, it is clear that they have provided a comprehensive rebuttal and taken diligent care in addressing all the questions.  Hence, I have decided to revise and increase my score accordingly.

---

### Official Review · Reviewer_3txU · 2023-07-12

**Soundness:** 3 good
**Presentation:** 3 good
**Contribution:** 4 excellent
**Rating:** 7
**Confidence:** 4

**Summary:**

This work focuses on

**Strengths:**

Originality: Related work in Chen et al., IEEE SMC 2022 uses a language model for pseudo label corrections during BCI self-recalibration, though the study is focused on simulations with EEG data from a longitudinal study with participants with ALS using the P300 speller. This work is a creative combination of existing ideas to develop a new method to recalibrate BCIs for communication by using language models to improve pseudolabel quality and enhancing continuous learning during recalibration via the use of a replay buffer and data augmentation.

Quality: This paper presents results from a longitudinal online BCI study to demonstrate utility of the proposed approach, which is the gold standard in evaluating BCI algorithms. The inclusion of results from offline analysis also enhances the paper.

Clarity: The paper is very well-written and organised. Areas needing clarity and suggestions to improve readability are noted below.

Significance: This work is highly relevant to developing automated approaches to periodically recalibrate BCIs for communication for long-term BCI use with minimal user disruptions.  The approach is applicable to general BCIs for communication. Results from a longitudinal online study with a BCI user from a target end user population increase the impact of the paper.


**Weaknesses:**

-	Low sample size. The paper presents results from one participant with generally high performance level, so difficult to assess the utility across a broad range of user performance levels.  The low sample size is understandable given the challenge with conducting studies in target BCI end-user populations; in particular, this is an iBCI study, in contrast to a non-invasive BCI study. The authors recognise the limitation of the lack of generalisability of results given the low sample size. The authors include results from simulations using data from the current participant to investigate the impact of a broad range of character error rates on the recalibration performance (Figure 5).

-	Potential order effects due to lack of randomisation of the no recalibration block (block 2) and the recalibration blocks (blocks 3 and 4). If understood correctly, the RNN decoder is updated with the data from the current calibration block and does not rely on data from the seed model block, so the testing order could be randomised daily to mitigate order effects.

-	There is the confound of the recalibration blocks (blocks 3 and 4) displaying the LM-decoded outputs (“the top-scored result was displayed on the screen as the final decoded sentence.”) vs. the no recalibration blocks (block 2) displaying the RNN-decoded outputs. This difference in feedback may potentially impact the BCI user experience (mental state, motivation, etc.) and further compound order effects as the user is aware given the fixed block order.


**Questions:**

-	“The second block employed a frozen seed model, trained on a combination of data from [46] and data collected prior to this evaluation (21 sessions in total).” Are the “data collected prior to this evaluation” from the current participant? Also referred to as “newly collected data” earlier in the paper. Are these 21 sessions prior to day 0?

- “updating the decoder after every sentence.” How is the end of a sentence detected? Automatically?

- Equation 1: \theta_k, where k denotes a day implies that the recalibration uses all the data from that day. Is Equation 1 supposed to be \theta_{x, k}, where x refers to sentence?

-	What are the character error rates of the LM-decoded outputs (Figure 2b)? This is to assess whether the use of LM-based correction at word level introduces errors at the character level (vs. Figure 2a with RNN-decoder outputs). (Can be inferred based on simulations in figure 5).

-	Figure 2: It would be useful to include results from recalibration with the ground truth labels. Why are the amounts of data collection different on day 0 and day 105 different? If understood correctly, day 0 does not have four blocks. This needs to be specified/clarified in the text/caption.

-	Inconsistency:  Over an approximately 8-month period, our participant used the iBCI system monthly and wrote on average 57.7 sentences per usage session.” vs. “X’s writing speed was 69.5 ± 8.6 characters per minute on average.”

-	What is “per-frame labeling”?

-	x_{i, t}, y_{i, t}: define subscript i.

-	Define all acronyms and variables in the captions and provide more context such that the captions standalone to understand the content of the presented information without necessarily referencing the text. Figure and table captions should be more informative to minimise confusion/misinterpreting the CER% or WER% results across figures/tables. For example, the mismatch between the average online WER % with CORP in table 1 vs. figure 2 is explained in the text and not the figure caption. Same with Figure 3. Captions should state if results are from offline vs. online analysis, specific blocks used during recalibration, etc., for clarity.

-	Check that the contrast between line styles is preserved when figures are in grayscale.


**Limitations:**

- More discussion is needed on the societal impact of the BCI technology. In particular, how effectively the BCI communicates a user’s intent when there is no alternative. There is the potential concern that the LM may be more dominant than the user’s intent, particularly in cases with low BCI prediction accuracy.

- “we do not anticipate pseudolabel quality to be a major concern in practice. This is because future clinically viable iBCIs are expected to have a high decoding accuracy...  Users are also likely to utilize the iBCI frequently, resulting in small nonstationarities most of the time. ... we believe that the pseudo-labels will have high accuracy, allowing CORP to sustain the iBCIs accuracy indefinitely.” Given the low sample sizes and no current data from “future clinically viable iBCIs”, these claims are questionable. There are issues related to recording quality with long term use of intracortical electrodes.

---

> ### Author Rebuttal · Authors · 2023-08-07
>
> > **Related work in Chen et al., IEEE SMC 2022 ...**
>
> Thank you for pointing out this related work. We’ll add it to the revised manuscript.
>
> > **Weakness 2**
>
> We appreciate this suggestion. We collected new data recently (two additional days - 278 and 300, shown in the attached pdf). On day 300, we ran recalibration blocks first followed by  the no-recalibration block. The results are still consistent with all other sessions. We’ll keep the session order randomized in all of our upcoming sessions and include this new data in the final paper.
>
> > **Weakness 3**
>
> It appears there was a lack of clarity in our methods description. Please note that both the recalibration blocks and the no-recalibration blocks display the LM-decoded outputs. We apologize for any confusion this may have caused, and we will make sure to clarify this point in the revised manuscript.
>
> > **Question 1**
>
> Yes, all the prior data is from the same participant.
> The 21st session is day 0. In this session, we collected 60 sentences, trained a decoder with 50 of those sentences combined with the previous 20 sessions’ data, and evaluated the decoder’s performance on the remaining 10 sentences. We’ll make this clear in the revised manuscript.
>
> > **Question 2**
>
> In this study, the participant would indicate to us when they had finished, and we would manually stop the decoder (since this particular participant can still speak, although he is paralyzed from the neck down). However, we believe the end of the sentence could automatically be detected without much difficulty, and this would be important for a locked-in user (the main target population of such a device)  One potential method is to use the output of the RNN itself. The RNN we used in this study is capable of outputting a special "blank" state when the participant is not writing. If the RNN outputs this blank state for more than a certain threshold of time (t seconds), we can infer that the participant has finished writing.
>
> > **Question 3**
>
> Yes, thank you for catching this error. It should be \theta_{j, k}, where j refers to a sentence sample of day k. We will update Eq 1 in the revised manuscript.
>
> > **Question 4**
>
> The average character error rate on LM-decoded outputs is 1.9% ± 0.6 (5.9% ± 1.4 for RNN-decoded outputs). The fact that LM can significantly reduce word error rate (6.3% ± 2.3 for LM-decoded outputs and 25.1% ± 5.6 for RNN-decoded outputs) means that using LM can reduce both word error rate and character error rate. It’s not shown in Figure 2 due to space limitations.
>
> > **Question 5**
>
> Thank you for this suggestion. The data used to plot Figure 2 was collected during online evaluation. Unfortunately, as we didn’t run any ground truth recalibration blocks online, we cannot include this as an additional online baseline. In the future, we will consider including such blocks for comparison as we agree they would be valuable.
>
> Day-0 was intended only to establish a baseline performance. As mentioned above, we collected 60 open-loop sentences in that session, and used the first 50 together with past data to train a RNN decoder and evaluated the decoder on the last 10 sentences. The performance shown in Figure 2 is on those 10 sentences.
>
> On day-105, we had some technical issues so only one recalibration block was collected.
>
> We’ll revise the manuscript to make this clear.
>
> > **Question 6**
>
> We would like to clarify that these  two statements refer to different metrics. The first statement refers to the total number of sentences written per session. The second statement refers to the rate of writing in characters per minute. These two metrics are distinct and provide different insights into the participant's usage and performance with the iBCI system. We hope this clarification is helpful, but please let us know if this does not resolve the issue.
>
> > **Question 7**
>
> Per-frame labeling means that each decoding frame (20ms time windows) needs to be assigned a ground-truth label. However, since our participant is tetraplegic, it’s impossible to get such labels. In [46], the authors used a hidden markov model to force align the neural data with the text to generate per-frame labels.
>
> > **Question 8**
>
> i indexes the trials in day t. We’ll revise the manuscript to include this.
>
> > **Question 9**
>
> Thank you for these helpful suggestions. We’ll revise the manuscript to include these clarifications.
>
> > **Question 10**
>
> Thanks for this suggestion. We’ll revise the manuscript to make sure that lines are distinguishable when in grayscale
>
> > **Limitation 1**
>
> We appreciate your feedback on the societal impact. We recognize the importance of accurately communicating a user's intent, and while handwriting iBCIs have shown high accuracy (>95% CER), a more comprehensive metric may be needed. We also acknowledge concerns about the LM dominating user intent, especially with low BCI accuracy, but note that this can be controlled via a weight parameter (Equation 3 in the Supplement). We will incorporate these points into the revised manuscript.
>
> > **Limitation 2**
>
> We acknowledge that the quality of pseudo-labels is indeed a crucial factor for the success of our proposed method. While we anticipate that future clinically viable iBCIs will have high decoding accuracy, we understand that this is a hypothesis that needs to be tested with more extensive data and over longer periods. Similarly, we recognize that there are potential issues related to the recording quality with long-term use of intracortical electrodes. This is an area that requires further investigation and we are committed to exploring this in our future work.
>
> We will revise our manuscript to more clearly acknowledge these concerns and the need for further research. We will also temper our claims to more accurately reflect the current state of knowledge and the limitations of our study.

---

> > ### Comment · Reviewer_3txU · 2023-08-14
> > **Post-rebuttal**
> >
> > Reviewer has read and appreciates the author rebuttal. The main concerns are mostly addressed. Revising score upward.
> >
> > Other: Suggest including example trajectories of characters, RNN-decoded and LM-decoded outputs.

---

### Official Review · Reviewer_miLw · 2023-07-26

**Soundness:** 3 good
**Presentation:** 4 excellent
**Contribution:** 4 excellent
**Rating:** 8
**Confidence:** 3

**Summary:**

Edit: I have read the rebuttal and am satisfied with the authors' comments. I stand by my score and would like to see this paper accepted.

########################################################################

In this work, the authors tackle the issue of non-stationarity in decoding technologies specifically for handwriting recognition. They propose several methodological steps to address like the use of a replay buffer, data augmentation and most importantly, an n-gram language model. They use data from a longitudinal study spanning over 200 days where they collect data from a single participant. Their model comprises an affine transform for each day followed by a 2-layer RNN that outputs character probabilities. These probabilities are then fed into a pertained n-gram language model that then generates plausible words (with beam search). The top-n words are then fed into GPT2-XL to rerank and the best word is displayed back to the participant. The output of the LM is also fed into a calibrator that uses this data along with some percentage of past data to recalibrate the RNN.

The authors show that this model achieves very good character error rate and reasonable word error rate over a model that doesn't recalibrate for stationarity and an alignment method. Through several experiments, they also show the usefulness of different model components, hyperparameter choices and the reliability of the pseudo-labels.

**Strengths:**

Strengths:
1. Very interesting applications of ML techniques to handwriting decoding.
2. Well written and easy to follow.
3. Achieves very good performance and authors clearly show through many experiments how their method alleviates the problem of non-stationarity.
4. Good analyses on model choice with ablations and many different parameter sets.

**Weaknesses:**

Weaknesses: No major weaknesses apart from some questions below and a few missing details.

**Questions:**

How long does the assumption of some stationarity hold? The authors say hat prior work found only a 1.5% error rate but is this something we can evaluate with respect to the drop in calibration error to find the most optimal window?

Re hyperparameter tuning in figure 4: It was not clear what the shaded region represented here- standard error across all decoded characters? Also, how stable is this across different days, ie., does the non-stationarity affect the plots substantially?

There were limited details provided on the affine transform for each day. How is the affine transform for each day trained and what role does it play in the online decoding system?

Clarification: Since the LM outputs word n-grams, I assume that this is segmented into characters to provide the pseudo `y's`?

Minor: What `p`was used in the final model? (esp for Table 4) Similarly, what is the `n` used for the GPT2-XL ranking step?

I would also encourage the authors to allude to the effect of replay buffer size and other modeling parameters on CER in the main text.

**Limitations:**

Yes, potential limitations were discussed.

---

> ### Author Rebuttal · Authors · 2023-08-07
>
> > **Question 1**
>
> Thank you for this insightful question regarding the assumption of stationarity and the optimal window for recalibration. The determination of the optimal window is a complex issue, primarily because the nature of nonstationarity in iBCI systems is unknown and can be highly unpredictable. Factors such as variability between individual participants, the recording modality used, and the specific tasks being performed can all influence the degree and pattern of nonstationarity. Given these complexities, a comprehensive analysis of this problem requires collaboration with more iBCI research labs and the collection of additional data. In our future work, we plan to pursue these collaborations to gain a deeper understanding of the underlying factors affecting nonstationarity and to develop more effective methods for managing it.
>
> > **Question 2**
>
> The shaded region in Figure 4 represents a confidence interval taken across 10 random seeds (we’ll update the figure caption to indicate this), computed via bootstrap resampling. Each data point in the figure is the average character error rate of all the recalibration sentences. This figure is to show that the recalibration process is stable around the operating points.
>
> Unfortunately we didn't compare these hyperparameters across different recalibration sessions to assess the effect of nonstationarity on these hyperparameters. However, given the stable recalibration accuracy in Figure 2 and the tight confidence interval around the operating point in Figure 4, we believe that nonstationarity has little influence on these hyperparameters. Thank you for bringing this to our attention, and we hope this explanation clarifies your query.
>
> > **Question 3**
>
> The affine transform is defined as:
>
> $y = Ax + b$, $x \in \mathbb{R}^{c \times 1}, A \in \mathbb{R}^{c \times c}, b \in \mathbb{R}^{c \times 1}$
> where $x$ is the input, $y$ is the transformed input, and $c$ is the input dimension.
>
> Each session day has its own affine transform layer. The affine transform layers are trained together with the RNN. For a new session, a new affine transform is created and its weights are initialized with the previous session’s. During online decoding, the input neural features are transformed by the affine layer first before being processed by the RNN.
>
> More details about it can be found in [46]. We’ll include the above in the revised manuscript.
>
> > **Question 4**
>
> Yes. LM outputs words, which are then converted into character-level pseudo-labels.
>
> > **Question 5**
>
> During online evaluation, we set p = 0.6, n = 100. We’ll add these to Table 4 in the Supplementary.
>
> > **Question 6**
>
> Since we only have a few hundred sentences of data in total, we loaded all the data into the replay buffer. During recalibration, the replay buffer samples BATCH_SIZE * p of sentences from the new data, and BATCH_SIZE * (1 - p) from the past data. We'll make this point clear in the revised manuscript.

---

### Author Rebuttal · Authors · 2023-08-07

We would like to express our sincere gratitude for your time and effort in reviewing our manuscript. Your constructive feedback and insightful questions have been invaluable in helping us improve the quality of our work.

We acknowledge the limitation of our study regarding the low sample size. Conducting studies with intracortical brain-computer interfaces (iBCIs) presents unique challenges, particularly in recruiting participants, compared to non-invasive BCIs. The nature of iBCIs, which involve invasive procedures, inherently limits the number of participants we can involve in our studies. We understand the implications of this limitation on the generalizability of our results and appreciate your understanding in this regard. In the future, we plan to address this limitation by collaborating with other iBCI labs to test our method on a broader range of subjects. Additionally, we plan to publish the code associated with our method. This will allow other researchers to test our method on their data, further contributing to the validation and generalizability of our findings.

The field is at an early stage in its  efforts to address nonstationarity in iBCIs. Our work represents an initial step towards developing a solution that can maintain the stability of iBCIs over extended periods. We hope that the encouraging results we have obtained so far and the publication of our dataset will stimulate interest in this problem among the broader machine learning community. We believe that the involvement of more researchers in this area will accelerate progress towards a robust solution to the nonstationarity problem in iBCIs.

Once again, we thank you for your thoughtful reviews and look forward to your continued feedback as we strive to improve our work.

An updated Figure 2 is attached to show the performance of CORP over a 300-day period, extending the original study from 228 days.

---

### Decision · Program_Chairs · 2023-09-21

**Decision:**

Accept (spotlight)

**Comment:**

The idea of using language-model input to help recalibrate brain-computer interfaces (BCIs) is interesting and well-motivated, and the methods in this submission are well-executed and promising.  While concerns were raised about the limitation to a single subject for the multi-month experimental evaluation, given the invasive nature of intracortical BCIs, most reviewers find this reasonable and still worthy of publication.  Please do carefully revise to clarify the various points discussed in the reviews and rebuttals.